# Circulating Tumor DNA and Management of Colorectal Cancer

**DOI:** 10.3390/cancers16010021

**Published:** 2023-12-19

**Authors:** Matthew Krell, Brent Llera, Zachary J. Brown

**Affiliations:** Department of Surgery, Division of Surgical Oncology, NYU Langone Health, NYU Grossman Long Island School of Medicine, Mineola, NY 11501, USA; matthew.krell@nyulangone.org (M.K.); brent.llera@nyulangone.org (B.L.)

**Keywords:** ctDNA, colorectal cancer, metastasis, liquid biopsy, biomarker

## Abstract

**Simple Summary:**

Colorectal cancer is a leading cause of cancer-related death in the United States. Liquid biopsies, such as the detection of circulating tumor DNA (ctDNA), have been investigated as a biomarker for patients with colorectal cancer in terms of prognosis and recurrence, as well as their use to guide therapy. CtDNA may provide additional information to assist in the decision for the administration of chemotherapy after surgery as well as the early detection of cancer recurrence. However, further information is needed in order to fully utilize ctDNA as a therapeutic biomarker.

**Abstract:**

Although the incidence of colorectal cancer (CRC) has decreased as a result of increased screening and awareness, it still remains a major cause of cancer-related death. Additionally, early detection of CRC recurrence by conventional means such as CT, endoscopy, and CEA has not translated into an improvement in survival. Liquid biopsies, such as the detection circulating tumor DNA (ctDNA), have been investigated as a biomarker for patients with CRC in terms of prognosis and recurrence, as well as their use to guide therapy. In this manuscript, we provide an overview of ctDNA as well as its utility in providing prognostic information, using it to guide therapy, and monitoring for recurrence in patients with CRC. In addition, we discuss the influence the site of disease may have on the ability to detect ctDNA in patients with metastatic CRC.

## 1. Introduction

Colorectal cancer (CRC) is the fourth leading cause of cancer and the second leading cause of cancer-related death in the United States (US), with over 100,000 cases of colon cancer and 45,000 cases of rectal cancer diagnosed annually [1]. As a result of increased screening efforts, the overall incidence of CRC has decreased. However, in patients less than 50 years old, the incidence of CRC has increased [1,2]. Metastatic disease is a leading cause of death in patients with CRC, and despite appropriate screening, 50–60% of patients eventually develop metastatic disease [3,4,5]. Additionally, 20 to 30% of patients present with synchronous liver metastases and 17% with peritoneal metastases (PM) [6,7,8,9]. Adjuvant chemotherapy has been demonstrated to improve disease-free survival (DFS) and overall survival (OS) in patients with localized resected CRC, but not all patients have been shown to benefit from adjuvant therapy [10,11]. In addition, there has been minimal evidence that intensive surveillance with clinical and endoscopic exams improved OS [12,13]. 

Many questions remain about the proper selection of patients and therapy so as to not overtreat patients at low risk of recurrence or undertreat those who are at a high recurrence risk. Biomarkers are of interest in many malignancies, including CRC, to offer prognostic information and help guide therapy [14,15]. For example, alfa-fetoprotein (AFP) has been used as a biomarker for hepatocellular carcinoma and is associated with aggressive tumor biology and high risk of tumor recurrence after potentially curative treatment such as liver transplantation [16]. Liquid biopsies, such as the detection of circulating tumor DNA (ctDNA), have been investigated as a biomarker for patients with CRC in terms of prognosis and recurrence, as well as their use to guide therapy. Herein, we provide an overview of ctDNA, as well as its utility in providing prognostic information, using it to guide therapy, and monitoring for recurrence. In addition, we discuss the influence the site of disease may have on the ability to detect ctDNA in patients with metastatic CRC.

## 2. Overview of Liquid Biopsies: ctDNA

Liquid biopsies involve the detection of tumor-derived biomarkers in patient fluid samples such as blood, urine, saliva, stool, cerebrospinal fluid, or bile. CtDNA or RNA, circulating tumor cells (CTC), and tumor derived exosomes, cytokines, and proteins are all of interest as liquid biopsy biomarkers [17,18] (Figure 1). Liquid biopsies can often be performed serially during a patient’s disease course and used to interpret disease biology, as well as changes in tumor biology and assess tumor response to therapy [18,19,20,21,22]. CtDNA is but a small fraction of circulating cell-free DNA (cfDNA) [23]. CtDNA, unlike cfDNA, is considered to be tumor-specific [24]. The ratio between ctDNA and cfDNA can greatly vary anywhere between 1 and 40% due to the characteristics of the primary tumor as well as the presence and location of metastatic disease [17,20,25].

Liquid biopsies and biomarkers, such as ctDNA, are gaining significant interest in the management of patients with colorectal cancer in terms of screening, prognosis, treatment monitoring, and treatment guidance. Different techniques for the detection of ctDNA include real-time quantitative polymerase chain reaction (qPCR), digital droplet PCR (ddPCR), and next-generation sequencing (NGS).

Multiple techniques, such as real-time quantitative polymerase chain reaction (qPCR), digital droplet PCR (ddPCR), and next-generation sequencing (NGS), have been utilized for the detection and analysis of ctDNA [26]. The detection of ctDNA can either be targeted or untargeted, while DNA may enter the bloodstream through both active and passive mechanisms [27,28,29,30]. The targeted approach for ctDNA detection identifies previously determined genetic mutations, while an untargeted approach does not require prior knowledge of underlying genetic alterations [27]. Kloten et al. identified KRAS mutations in patients with CRC using Intplex PCR and ctDNA analysis, which demonstrated a 70% specificity and 50% sensitivity when compared to serum samples, with the tissue samples having a concordance of 66% [31]. Tumor DNA may enter the bloodstream due to migration from the primary tumor or metastatic site by tumor invasion, shedding, or mechanical stress [29,30]. Passive mechanisms include the release of DNA remnants through apoptosis and necrosis. Meanwhile, active mechanisms are not completely understood but may be due to the excretion of DNA through micro-vesicles containing fragments of ctDNA [29].

CfDNA has been found to be significantly higher in cancer patients as compared to healthy individuals, but serum levels may be influenced by a variety of factors [24]. Compared to healthy controls, CRC patients have higher levels of mutated DNA in their bloodstream as well as higher levels compared to other solid organ malignancies [32,33,34]. Additionally, ctDNA detection is also influenced by the stage of the disease. CtDNA has been shown to be detectable in approximately 46% of patients with stage I disease, 73% of patients with stages II–III, and 90% of patients with metastatic CRC [32,35]. As a result, there is considerable interest in utilizing ctDNA as a screening method for CRC in patients who are not willing to undergo colonoscopy [36]. On the other hand, carcinoembryonic antigen (CEA) has been the conventional tumor marker for evaluating patients with CRC for recurrence and therapeutic efficacy. Osumi et al. evaluated 110 patients with metastatic CRC undergoing chemotherapy to evaluate the correlation between ctDNA and CEA. The overall concordance rate between the ctDNA and CEA levels was 75.5% (83/110). Additionally, the correlation coefficient between ctDNA and CEA levels was lower in patients without liver and lymph node metastases (r = 0.18, *p* = 0.44) than in patients with liver metastasis (r = 0.48, *p* < 0.0001) [37].

## 3. Influence of Site of Metastasis on ctDNA

A majority of patients with CRC eventually develop metastatic disease. The influence of the site of metastatic disease has been investigated by its correlation with ctDNA. Bando et al. investigated the relationship between the site of metastasis and ctDNA in patients with single-organ CRC metastasis as part of the SCRUM-Japan GOZILA study. Of the 1187 patients with metastatic CRC enrolled in GOZILA, 138 patients were studied: 49 with liver metastasis, 15 with lymph node metastasis, 27 with peritoneal metastasis (PM), and 47 with lung metastasis. In this study, patients with lung metastases and PMs had significantly lower levels of ctDNA [38]. Similarly, in a retrospective analysis, Sullivan et al. identified patients with stage II-IV GI cancers treated at a single institution between 2015 and 2020 with available ctDNA results. PMs were associated with lower ctDNA levels independent of the primary tumor site (PM only: 12.1%; PM with visceral metastases: 26.8%; and visceral metastases only: 35.0%; *p* < 0.01) [39]. The authors hypothesized that PMs were associated with lower ctDNA levels than other metastatic sites due to the plasma-peritoneal barrier. Xue et al. performed a systematic review on the utility of ctDNA in CRC PM, including eight studies with 167 patients. The authors not only found that ctDNA can be isolated from both plasma and peritoneal fluid, but peritoneal fluid had higher mutation detection rates and was preferred for liquid biopsy [40]. Furthermore, as compared to patients with CRC PM, patients with CRC liver metastases have higher detectable ctDNA. Lee et al. investigated patients who underwent metastasectomies for metastatic CRC, where the detection rate of ctDNA was higher in patients with liver metastasis and tumors measuring ≥1 cm [41]. Therefore, more information is required to understand the utilization of ctDNA in the treatment of metastatic CRC, as different metastatic disease sites may influence ctDNA levels. 

## 4. Prognostic Impact of ctDNA

The ability to provide prognostic information on recurrence and survival has been the backbone of cancer care. For example, in patients with melanoma, the use of a sentinel lymph node biopsy does not offer therapeutic benefits to patients but rather provides proper staging and diagnostic information to identify patients at high risk of recurrence where additional therapies may offer survival benefit [42,43]. Post-operative detection of ctDNA in patients with CRC has been found to be an independent marker of disease recurrence. In addition, compared to CT and CEA, ctDNA has a lead time of approximately 10 months [44]. In patients with stages I to III CRC, ctDNA detection 30 days post-operatively was associated with a higher likelihood of recurrence than patients who did not have detectable ctDNA (HR 72; 95% CI, 2.7–19.0, *p* < 0.001) [44]. Similarly, 150 patients with localized colon cancer were prospectively evaluated for ctDNA via NGS following surgery [45]. Again, detection of ctDNA was associated with poor DFS (HR, 17.56; log rank *p* = 0.0014). Furthermore, data indicate that patients with positive ctDNA prior to surgery not only have a higher risk of recurrence but also a shorter time to recurrence [46]. In addition to patients with positive ctDNA having worse outcomes after surgical resection, similar results are seen in patients with positive ctDNA after adjuvant chemotherapy. Reinert et al. found patients with positive ctDNA after adjuvant chemotherapy were 17 times more likely to have disease recurrence (HR 17.5; 95% CI, 54–56.5; *p* < 0.001). Additionally, ctDNA analysis found disease recurrence up to 16.5 months ahead of standard of care radiologic imaging (mean 8.7 months; range: 0.8–16.5 months) [47]. 

The use of ctDNA was also studied in patients with locally advanced rectal cancer. In 159 patients with locally advanced rectal cancer, Tie et al. measured plasma ctDNA pre-treatment, after chemoradiation, and after surgical resection. The three-year recurrence-free survival (RFS) was 33% for patients with detectable ctDNA vs. 87% for patients with no detectable ctDNA after surgery [48]. In a study by Vidal et al. of patients with locally advanced rectal cancer treated with fluorouracil, leucovorin, and oxaliplatin (mFOLFOX6) with or without aflibercept, followed by chemoradiation and surgery, ctDNA was detectable in 83% of patients at baseline and in 15% following total neoadjuvant therapy (TNT). There was found to be no association between ctDNA status and the pathologic response to TNT. However, detectable pre-operative ctDNA was significantly associated with systemic recurrence, shorter DFS (HR, 4; *p* = 0.033), and shorter OS (HR, 23; *p* < 0.0001) [49].

Kotani et al. reported data from the GALAXY study, which is the observational arm of the ongoing CIRCULATE-Japan study (UMIN000039205) that analyzed presurgical and postsurgical ctDNA in patients with stage II–IV resectable CRC. In this study, postoperative ctDNA positivity was associated with higher recurrence risk (HR 10.0, *p* < 0.0001) and was the most significant prognostic factor associated with recurrence risk in patients with stage II or III CRC (HR 10.82, *p* < 0.001). Additionally, postoperative positive ctDNA identified patients with stage II or III CRC who received benefit from adjuvant chemotherapy (HR 6.59, *p* < 0.0001) [50]. Similarly, Jones et al. performed a meta-analysis of 28 studies analyzing ctDNA in stage IV CRC patients, finding a strong correlation between measurable ctDNA after treatment with OS (HR 2.2, 95% CI 1.79–2.69, *p* < 0.00001) and progression-free survival (PFS) (HR 3.15, 95% CI 2.10–4.73, *p* < 0.00001). Furthermore, in patients with resectable disease treated with curative intent, detection of ctDNA offered a lead time of 10 months over radiological recurrence [51]. In patients with metastatic disease receiving chemotherapy, there was a high correlation between the ctDNA response and median survival [52]. In patients with metastatic CRC, Jia et al. examined serial changes of ctDNA in patients receiving first-line chemotherapy found that ctDNA reductions as early as prior to cycle 2 predicted responses after cycle 4 [53].

The treatment of metastatic CRC has evolved, with patients undergoing locoregional therapies for metastatic disease. Liver transplantation is now an accepted treatment strategy in well-selected patients with liver-confined, unresectable CRC liver metastasis. The utility of ctDNA in this population was assessed before (4 patients) and after transplant (6 patients). They found that four patients were negative for ctDNA following transplant, and two patients had persistently positive ctDNA after transplant. Three of four patients with positive ctDNA before transplant are ctDNA negative after transplant [54]. However, more studies are required to investigate ctDNA in the setting of liver transplantation.

Cytoreductive surgery with or without hyperthermic intraperitoneal chemotherapy (CRS/HIPEC) is a therapeutic option for select patients with CRC PM. Beagan et al. explored the possibility of utilizing ctDNA to assist in selecting patients for CRS/HIPEC. Thirty patients eligible for CRS/HIPEC provided blood samples preoperatively and during follow-up if the procedure was completed. CtDNA was detected preoperatively in cfDNA samples from 33% of patients and was associated with a reduced disease-free survival (DFS) after CRS/HIPEC (median 6.0 months vs. median not reached, *p* = 0.016) [55]. Similarly, Dhiman et al. evaluated patients with CRC or appendiceal cancer with PM who underwent CRS/HIPEC and had post-resection ctDNA monitoring. One hundred thirty serial post-resection ctDNA assessments were performed in 33 patients who underwent complete CRS. Of the 19 patients with rising ctDNA levels, 90% recurred vs. 21% in the stable ctDNA group (*n* = 14, *p* < 0.001). Median DFS in the rising ctDNA cohort was 11 months and not reached in the stable group (*p* = 0.01). A rising ctDNA level was the most significant factor associated with DFS (HR: 3.67, 95% CI, 1.06–12.66, *p* = 0.03) [56].

## 5. Application of ctDNA to Guide Treatment

Traditionally, patients with stage II colon cancer do not receive adjuvant chemotherapy, while the vast majority of patients with stage III disease are recommended to receive three to six months of adjuvant chemotherapy [57]. Several studies have been conducted investigating the role of ctDNA to guide adjuvant treatment. The DYNAMIC study evaluated 455 patients with a ctDNA-guided approach to adjuvant chemotherapy. Patients with stage II CRC were randomly assigned to have treatment decisions guided by either ctDNA results or standard of care. In patients assigned to the ctDNA-guided arm, adjuvant chemotherapy was administered to patients with positive ctDNA 4 or 7 weeks after surgery. As a result, a lower percentage of patients in the ctDNA-guided arm received adjuvant chemotherapy. In addition, at 2-year follow-up, the RFS in the ctDNA-guided arm was non-inferior to the standard administration of adjuvant chemotherapy [58]. Furthermore, cost–utility analysis found a ctDNA-guided strategy increased quality-adjusted life-years and was shown to be a potentially cost-effective strategy towards reducing overtreatment in stage II colorectal cancer [59].

Traditional biopsies provide a static analysis on a tumor at a given time and location rather than providing information of tumor heterogeneity and changes occurring over time. In addition, there can be differences between the primary tumor and metastatic lesions, as tumors are often comprised of different clones [60,61]. CtDNA sequencing has allowed for repeated tumor mutational profiling through a noninvasive means. In a study by Kim et al., new mutations in samples from patients with progressive CRC were seen in 49.6% of treatments [62]. Additionally, patients with tumors that were RAS/BRAF wild-type were more likely to develop mutations causing progressive disease independent of anti-EGFR treatment with cetuximab [62].

Chemotherapy with anti-EGFR agents is standard first-line therapy in RAS-wild-type metastatic CRC. After treatment, the potential for conversion of RAS mutational status from RAS mutant to RAS wild-type has emerged, thus the concept of ‘NeoRAS wild-type’ [63]. RAS/BRAF mutations have been found to be associated with disease resistance detected after therapy, and an increase in RAS/BRAF ctDNA was associated with a shorter PFS [64]. Other studies have investigated the re-determination of the *RAS* mutational status in ctDNA at disease progression in RAS mutant metastatic CRC. The most commonly detected actionable mutations in “neo-*RAS* wild-type” were: *PIK3CA* (35.7%), *RET* (11.9%), *IDH1* (9.5%), *KIT* (7%), *EGFR* (7%), *MET* (4.7%), *ERBB2* (4.7%), and *FGFR3*(4.7%). OS and PFS were longer in patients with “neo-*RAS* wild-type” compared to those who remained *RAS* mutant [65]. Ciardiello et al. performed a further analysis of the CAVE and VELO studies, evaluating the percentage of patients with wild-type ctDNA tumors and the association of mutational status with time since the last anti-EGFR drug administration. The authors found no difference in the proportion of patients whose baseline plasma ctDNA was *RAS*/*BRAF* wild-type or mutated between 4 and 18 months since the last administration of anti-EGFR drugs. However, 38/44 patients with an anti-EGFR drug-free interval of 18 months or more had a ctDNA *RAS*/*BRAF* wild-type status, which supports the role of ctDNA assessment in improving treatment efficacy as the length of the anti-EGFR free interval is not sufficient for patient selection for further treatment [66]. Further investigation of RAS status in ctDNA as a useful biomarker of EGFR inhibitors for NeoRAS wild-type metastatic CRC is being explored (C-PROWESS trial) [63]. 

## 6. Treatment Monitoring

Detection of ctDNA may offer prognostic information in patients with CRC and identify patients at high risk for disease recurrence. Early detection of CRC recurrence by conventional means such as CT, endoscopy, and CEA has not translated into an improvement in survival [12,13]. CtDNA has been studied to investigate the recurrence of CRC and has been found to be more frequently positive than CEA (85% vs. 41%; *p* = 0.002) in CRC patients with disease recurrence seen on imaging. In addition, the time between ctDNA detection and radiologic disease recurrence has been found to be significantly shorter for ctDNA. This is also true when comparing ctDNA and CEA, leading to earlier detection (61 vs. 167 days, *p* = 0.04; ctDNA vs. CEA, respectively) [67]. Another study demonstrated that the lead time of ctDNA is approximately 8 months as compared to CEA [13]. Additionally, serial liquid biopsies allow response prediction prior to that obtained by conventional methods in patients with metastatic disease undergoing chemotherapy [68] (Figure 2). In a study by Zou et al., ctDNA was compared to CEA for the assessment of chemotherapy response in patients with metastatic CRC. In the study of thirty patients, 29 (97%) had detectable ctDNA, compared to 25 patients (83%) who were CEA positive. CtDNA not only predicted significantly more disease progression than CEA (16 (80%) vs. 6 (30%), respectively; *p* = 0.004) but also the rise in ctDNA occurred significantly earlier than CEA (*p* = 0.046) [69].

Serial liquid biopsies allow response prediction prior to that obtained by conventional methods in patients with metastatic disease undergoing chemotherapy. CtDNA may also prove useful in treatment monitoring of acquired resistance mechanisms and thus change treatment strategy.

CtDNA may also prove useful in treatment monitoring of acquired resistance mechanisms, combinatorial strategies to delay resistance, and the potential for anti-EGFR rechallenge as acquired mutations appear as multiple concurrent subclonal alterations [70]. Several studies have demonstrated that repeated determination of mutational status in RAS/BRAF wild-type patients or patients who acquired resistance during the course of treatment had improvements in PFS and OS [71,72]. Nakamura et al. conducted a phase 2 trial to evaluate pertuzumab plus trastuzumab for metastatic CRC with human epidermal growth factor receptor 2 (HER2) amplification confirmed by tumor tissue biopsy or ctDNA. The authors found that decreased ctDNA three weeks after treatment was associated with a therapeutic response. Additionally, duel Her-2 blockade with pertuzumab and trastuzumab demonstrated similar efficacy with HER2 amplification in tissue or ctDNA, demonstrating that ctDNA genotyping can identify patients who benefit from dual-HER2 blockade as well as monitor treatment response [73].

The current NCCN guidelines highlight that ctDNA testing is not without drawbacks, where a positive result without evidence of disease is only helpful if the patient has therapeutic options with a reasonable chance to eradicate disease. Early knowledge of cancer recurrences without effective interventions could lead to significant distress. More details are needed on the timing of the assay and the value quantification of ctDNA in ongoing studies.

## 7. Future Directions

CtDNA has emerged as a useful adjunct in the treatment and screening of patients with CRC. However, the full extent of its utility has not been fully elucidated, with potential for a wider array of its use in guiding prognosis and therapeutic interventions. For example, a rise in ctDNA has been shown to precede radiologic recurrence or an increase in CEA in patients off therapy undergoing surveillance. Yet, currently, there are no high-level data to justify restarting treatment, such as chemotherapy, based on a rise in ctDNA alone. This may prove beneficial in treating patients before gross disease recurrence. However, one must consider potential for overtreatment and the ability to act upon the information. Additionally, as discussed, differences have been found in ctDNA elevation based on recurrence patterns in the lung, liver, and peritoneum. In patients at high risk for peritoneal recurrence, such as patients with perforated tumors or T4 tumors, ctDNA may not be an ideal monitoring modality. Additional studies are required and ongoing for the evaluation of the use of ctDNA in patients with metastatic disease (Table 1). 

Recently, four molecular subtypes of CRC have been identified, each of which has significant biological differences and thus the potential for subtype-based intervention [74]. The CMS1 subtype makes up 14% of cases and is microsatellite unstable with strong immune activation; CMS2 constitutes 37% of cases with marked WNT and MYC signaling activation; CMS3 (13% of cases) has metabolic dysregulation; and CMS4 (23% of cases) demonstrates prominent transforming growth factor–β (TGF-β) activation with stromal invasion and angiogenesis [74]. However, several limitations exist in transforming these discoveries into clinical practice, including the ability to classify tumors, as classification is reliant on gene expression profiles with associated costs, time, and expertise. A potential solution may be in the subtype-specific liquid biomarkers to classify patients, thus the potential to change therapeutic strategy [75].

## 8. Conclusions

Circulating tumor DNA has been studied in its application as a biomarker for patients with CRC in terms of prognosis and recurrence, as well as its use to guide therapy. Elevated ctDNA following surgical resection or systemic chemotherapy has been found to be associated with a poor prognosis. The additional use of ctDNA could also lead to an improvement in the distribution of healthcare resources by identifying patients more likely to respond to therapy as well as alternative and specific treatment regimens. CtDNA may prove to be beneficial in guiding adjuvant therapy in patients with a high-risk stage II or low risk stage III disease. We expect the use of ctDNA in CRC to gradually change the management and therapeutic options for eligible patients, and we believe its use will eventually be broadened to cancers of other organ systems.

## Figures and Tables

**Figure 1 cancers-16-00021-f001:**
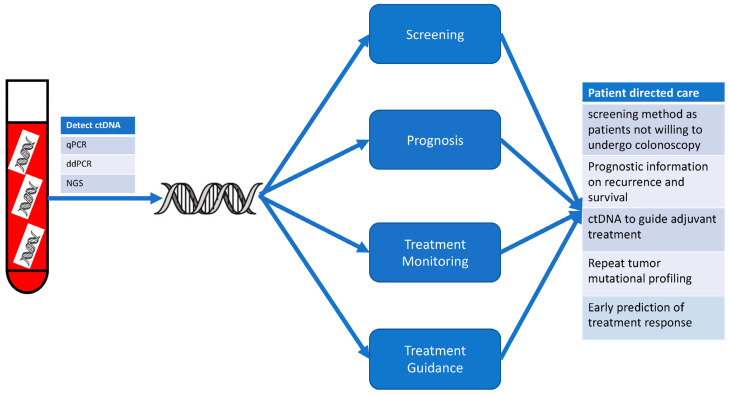
Utility of circulating tumor DNA in the treatment of colorectal cancer.

**Figure 2 cancers-16-00021-f002:**
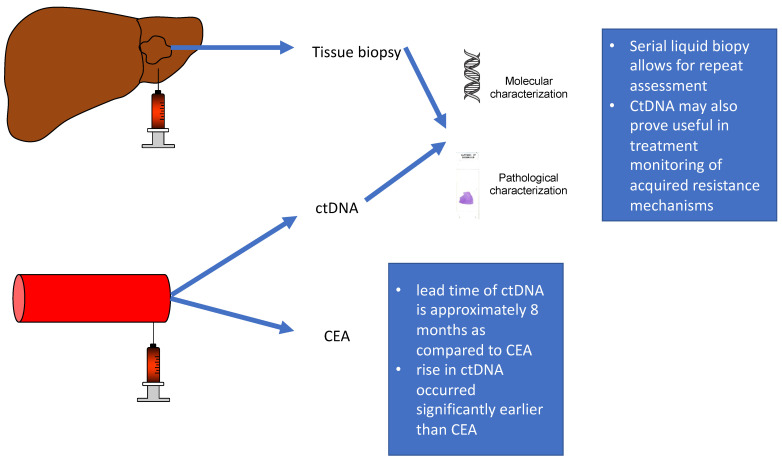
Drawing comparing tissue biopsy and serum liquid biopsy, ctDNA.

**Table 1 cancers-16-00021-t001:** Ongoing Studies of ctDNA in the Management of Metastatic Colorectal Cancer.

NCT Number	Title	Intervention	Summary	Outcomes
NCT01212510 Coca-Colon	Study of Circulating Markers in Serum of Patients Treated for Metastatic Colorectal Cancer	ctDNA	Evaluate the value of circulating free mutant DNA and circulating tumor cells and their variations during the treatment.	Prediction of tumor progression, kinetics to predict tumor progression at 3 months
NCT04466267	The Molecular Mechanism of RAS Wild-type mCRC Resistance to Anti-EGFR-antibody	ctDNA	Combination genetic data with clinical characteristics, prognosis and treatment data to explore the molecular mechanism of resistance of anti-EGFR-antibody.	The molecular mechanism of patients’ primary or secondary resistance to cetuximab with monitoring ctDNA to identify potential molecular mechanism of RAS wild-type mCRC patients’ primary or secondary resistance to cetuximab
NCT05755672 On-CALL	On-treatment Biomarkers in Metastatic Colorectal Cancer for Life	Resection of the primary tumor and metastases ctDNA sampling during chemotherapy	Generate further knowledge on the evolutionary progression of mCRC during treatment, and to elucidate the mechanisms underlying the therapeutic failure still seen in a substantial number of patients.	Radiological/clinical examination of tumor remission, progression or recurrence, and correlation of this clinical information with the available oncogenetic data from histological samples from the primary tumor and metastases, and from ctDNA analysis
NCT05635630	Predictive Value of ctDNA for NED Status in mCRC and Its Utility in Guiding Therapeutic Intervention	ctDNA and adjuvant therapy	The goal is to detect the prognostic value of longitudinal monitoring ctDNA for NED status in metastatic colorectal cancer patients and its utility in guiding therapeutic intervention.	RFS, detect the RFS time in mCRC patients with NED status who received ctDNA guided therapies. From date of surgery until the date of first documented progression or date of death from any cause, whichever came first, assessed up to 2 years.
NCT04752930	ctDNA as an Assisted Diagnosis, Early Intervention and Prognostic Marker for Peritoneal Metastases From Colorectal Cancer	ctDNA	Monitoring the serum ctDNA mutational profile using NGS, aiming to elucidate the correlation between the postoperative ctDNA status and the assisted diagnosis, early intervention and prognosis for colorectal cancer peritoneal metastases.	PMFS, The survival rate without peritoneal metastasis at 24 months after radical resection of CRC. DFS between ctDNA-positive patients treated with additional treatment of FOLFIRI and ctDNA-positive patients who are untreated Compare the clearance rate of ctDNA in ctDNA-positive patients between patients treated with additional treatment of FOLFIRI and those who are untreated
NCT05398380 METLIVER	Liver Transplantation for Non-resectable Colorectal Liver Metastases: Translational Research	Liver transplantation ctDNA	LT is a potential for patients without extrahepatic involvement and nonresectable CRLM. There are several studies that aims to evaluate if LT increases overall survival compared to best alternative care. No studies incorporate objectives focused on the underlying tumor biology of this particular population and the development of focused strategies including a dynamic disease monitoring and targeted treatments for this particular population.	Five years overall survival Percentage of subjects who reach the endpoint of overall survival from the inclusion in waiting list until death or last follow-up
NCT01983098	Analysis of Circulating Tumor DNA to Monitor mCRC Treatment	ctDNA	Compare the monitoring of circulating tumor DNA with the results of CT scan according the RECIST criteria and the blood level of CEA and CA 19-9	To compare the monitoring of ctDNA with the results of CT scan from first biomarker date to first clinical event.
NCT01212510	Study of Circulating Markers in Serum of Patients Treated for mCRC	ctDNA	Usefulness of the serum CEA kinetic for chemotherapy monitoring in patients with unresectable mCRC The secondary purpose is to evaluate the value of ctDNA and CTC and their variations during the treatment.	Prediction of tumor progression, sensitivity and specificity of CEA kinetic to predict tumor progression at 3 months
NCT04466267	The Molecular Mechanism of RAS Wild-type mCRC Resistance to Anti-EGFR-antibody	Anti-EGFR-antibody ctDNA	Combination genetic data with clinical characteristics, prognosis and treatment data to explore the molecular mechanism of resistance of anti-EGFR-antibody.	The molecular mechanism of patients’ primary or secondary resistance to cetuximab. Dynamic monitoring ctDNA to identify potential molecular mechanism of RAS wild-type mCRC patients primary or secondary resistance to cetuximab

ctDNA, circulating tumor DNA; mCRC, metastatic colorectal cancer; EGFR, epidermal growth factor receptor; NED, no evidence of disease; RFS, Relapse-free survival; PMFS, Peritoneal Metastasis Free Survival; DFS, disease free survival; LT, Liver transplantation; CEA, Carcinoembryonic antigen; CTC, circulating tumor cells.

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
