# Peer review of "Circulating Tumor DNA and Management of Colorectal Cancer"

_cancers, 2023, doi:10.3390/cancers16010021_

Round 1
Reviewer 1 Report
Comments and Suggestions for Authors
Krell et al described the Circulating Tumor DNA and Management of Colorectal Cancer. This review article is a recent collection of various approaches in ctDNA and its utility in providing prognostic information, guiding the therapy, and monitoring aspects. Authors also described the influence the site of disease on the ability to detect ctDNA in patients with metastatic CRC. This review is presented in a well organized way, however, some plagiarised texts need to reframed, accordingly.
1. 3rd and 4th paragraphs of introduction section are slightly plagiarised.
2. Influence of Site of Metastasis on ctDNA section also had plagiarised text.
3. The paragraph next to Figure 2 has plagiarised text.
Author Response
1. 3rd and 4th paragraphs of introduction section are slightly plagiarised.
Thank you for your comments- the manuscript was changed accordingly.
2. Influence of Site of Metastasis on ctDNA section also had plagiarised text.
The manuscript was changed accordingly.
3. The paragraph next to Figure 2 has plagiarised text.
The manuscript was changed accordingly.
Please note that as this is a review some text cannot be changed that contains significant data but is cited accordingly.
Reviewer 2 Report
Comments and Suggestions for Authors
Dr Krell and colleagues' work was a thorough review of the current knowledge about liquid biopsies in the context of colorectal cancer, as a prognostic tool, recurrence monitor, and therapy guide. The authors provided an overview of ctDNA and its utility in providing prognostic information, guiding therapy, and monitoring for recurrence in patients with CRC. They discussed the influence of the site of disease on the ability to detect ctDNA in patients with metastatic CRC. Additionally, they highlighted the potential of ctDNA in changing the management and therapeutic options for eligible patients and broadening its use to cancers of other organ systems.
The authors should be congratulated for this excellent work in this important topic. Here are some comments to improve the manuscript further:
Minor Revisions
-
Some typographical errors and inconsistencies in referencing (e.g., page 9, line 305).
-
Clarity issues in certain sections, particularly in the discussion of the influence of the site of disease on the ability to detect ctDNA (e.g., page 9, line 17).
Major Revisions
-
The manuscript would benefit from a more detailed discussion of the Consensus Molecular Subtypes and their relevance to liquid biopsies in CRC.
-
Additional data and statistical analysis could strengthen the conclusions drawn from the review (to be included in the main body of the manuscript).
Again, I enjoyed reading this manuscript.
Author Response
Minor Revisions
Some typographical errors and inconsistencies in referencing (e.g., page 9, line 305).
-Thank you for your comment and article was changed accordingly
Clarity issues in certain sections, particularly in the discussion of the influence of the site of disease on the ability to detect ctDNA (e.g., page 9, line 17).
-Additional clarification added accordingly to clarify.
The manuscript would benefit from a more detailed discussion of the Consensus Molecular Subtypes and their relevance to liquid biopsies in CRC.
-Thank you for the suggestion- this was added to the future directions section
Additional data and statistical analysis could strengthen the conclusions drawn from the review (to be included in the main body of the manuscript).
-This was added where appropriate and more data was added for molecular classification.